# Wearable Medical Devices: Application Status and Prospects

**DOI:** 10.3390/mi16040394

**Published:** 2025-03-28

**Authors:** Xiaowen Wang, Yingnan Gao, Yueze Yuan, Yaping Wang, Anqin Liu, Sen Jia, Wenguang Yang

**Affiliations:** 1School of Mechanical and Electrical Engineering, Yantai Institute of Technology, Yantai 264005, China; 17865570339@163.com (Y.G.); 18863893728@163.com (Y.Y.); wyp1944@126.com (Y.W.); liuan_821@163.com (A.L.); 2School of Electromechanical and Automotive Engineering, Yantai University, Yantai 264005, China; yangwenguang@ytu.edu.cn

**Keywords:** E-skin, health electronic device, communication technology, power supply

## Abstract

Electronic skin (E-skin) refers to a portable medical or health electronic device that can be worn directly on the human body and can carry out perception, recording, analysis, regulation, intervention and even treatment of diseases or maintenance of health status through software support. Its main features include wearability, real-time monitoring, convenience, etc. E-skin is convenient for users to wear for a long time and continuously monitors the user’s physiological health data (such as heart rate, blood pressure, blood glucose, etc.) in real time. Health monitoring can be performed anytime and anywhere without frequent visits to hospitals or clinics. E-skin integrates multiple sensors and intelligent algorithms to automatically analyze data and provide health advice and early warning. It has broad application prospects in the medical field. With the increasing demand for E-skin, the development of multifunctional integrated E-skin with low power consumption and even autonomous energy has become a common goal of many researchers. This paper outlines the latest progress in the application of E-skin in physiological monitoring, disease treatment, human–computer interaction and other fields. The existing problems and development prospects in this field are presented.

## 1. Introduction

Flexible sensors, also known as electronic skin (E-skin), provide a bionic perception of human skin by converting external information into electrical signals [1,2,3,4,5]. It has the comprehensive advantages of flexibility, light weight and user comfort, and has attracted great attention in the fields of intelligent robots [6,7,8,9], human–computer interaction [10,11,12,13,14,15], healthcare monitoring [3,16,17,18] and wearable electronic devices [19,20]. In recent years, the development of robotic technology has enabled E-skin to have excellent biocompatibility and mechanical properties, making it suitable for a variety of medical applications, such as disease diagnosis [21,22,23,24] and treatment, physiological monitoring, auxiliary robots and prostheses [25,26,27,28,29,30]. E-skin has similar durability and stretchability to human skin and has the ability to measure pH, blood glucose, temperature and pressure. In addition, by integrating excellent bioelectronic materials and devices, E-skin can have functions beyond normal human skin.

The key technologies of E-skin include sensor technology [31,32,33] (motion sensor [34,35,36,37], biosensor [38,39,40,41,42] and environmental sensor [43,44]) and medical chip technology for collecting users’ physiological health data, communication technology (Wi-Fi [19,45,46,47], Bluetooth [31,48,49,50], ZigBee [51,52,53], etc.) for transmitting collected data to mobile terminals or clouds for processing and analysis, and energy acquisition and storage technology (lithium battery or solar energy, mechanical energy and other environmental energy) to ensure that the equipment can work stably for a long time [5,54,55,56,57,58,59,60]. Battery is the main source of power supply for most wireless E-skin devices. However, the battery life of existing devices is limited. E-skin, which obtains energy from human motion, body heat and solar energy, is inefficient, so it is not enough to provide power for signal processing modules and wireless data transmission modules. In order to overcome the above problems, a battery-free E-skin for near-field communication (NFC) has been developed for wireless power supply and data transmission. However, the error of data processing and reading is large. Human sweat and blood are rich in chemical substances, which can not only reflect human health, but also provide sustainable power supply through chemical reactions. It is an ideal and sustainable bioenergy.

In this paper, the main manufacturing materials of E-skin are introduced in detail, and the latest progress of its application in physiological monitoring, disease treatment, human–computer interaction and other fields is summarized (Figure 1). The existing problems and development prospects in this field are presented.

## 2. Materials

E-skin mimics the ability of human skin to perceive various stimuli, focusing on sensing functions such as pressure, temperature, deformation and humidity, which is widely used in the field of medical monitoring and human–computer interaction. In summary, in order to effectively monitor these signals, the construction of flexible sensors indispensably includes three core components, the sensing element, the support structure and the conductive electrode, which provides a clear guide for material selection and mechanism design (Table 1). Specifically, the substrate shapes the mechanical and physical properties of the device, the sensing element determines its electrical properties, and the conductive electrode ensures the efficiency of electron conduction and data transfer.

### 2.1. Substrate Material

#### 2.1.1. High-Molecular Polymer

In general, the sensing function depends heavily on the selection and arrangement of materials. For example, flexible strain sensors are required to be able to adapt to the human body and can sense stretching, compression, twisting and bending, so they are suitable for disease diagnosis, rehabilitation and so on [12,61]. Therefore, choosing the appropriate substrate material is the key. The substrate material of E-skin requires flexibility, scalability, high sensitivity, good fit and comfort. First of all, the substrate material needs to be able to adapt to different parts of the human body. Even with various bending and stretching actions, it will not cause the material to crack or damage. Therefore, the substrate material needs to have good flexibility to ensure that the E-skin can adhere to the human body surface long-term and conduct stable health monitoring. Secondly, in order to capture tiny physiological signals quickly and accurately, the substrate material needs to have high sensitivity. In addition, comfort is also an important consideration for substrate materials. The substrate material should be as thin and breathable as possible to reduce irritation and discomfort to the skin.

Polyethylene terephthalate (PET) [24,62,63,64], polyimide (PI) [65,66] and other polymers are used as substrate materials due to their high strength and high interface stability. In addition, elastomeric polymer materials such as polydimethylsiloxane (PDMS) and polyacrylate can well simulate the characteristics of low mechanical modulus of human skin, and the substrate materials made from them will not break or be damaged due to excessive stretching or compression [62]. Among them, PDMS has low mechanical modulus, high strength and high interface stability, as well as good biocompatibility and transparency [62]. In particular, the distinct characteristics of the adhesive and non-adhesive regions under ultraviolet light make the surface easy to adhere to the electronic material. Therefore, PDMS is the most commonly used material for substrate fabrication.

Bending of human joints, such as the wrist, often causes human skin to separate from non-sticky wearable devices, which limits signal acquisition and long-term movement monitoring. In view of this, Dai et al. made an adherent E-skin based on PDMS, which consists of a bonding layer and a response layer, as shown in Figure 1B [67]. Among these, the adhesive layer plays a fixed role to ensure conformal coverage of the interface with the skin, thus maintaining the stability and accuracy of the signal. The microcilia of the response layer are mainly used for the perception of inward/outward curvature. In addition, Wang et al. assembled silver nanowires (AgNWs) and PET films with a mixed shear-reinforced polymer/polydimethylsiloxane (SST/PDMS) matrix to develop a novel multifunctional E-skin with protective and multi-sensing properties (Figure 1A). The hybrid SST/PDMS polymer is structurally stable. When the shear frequency varies from 0.1 to 100 Hz, the storage modulus of SST/PDMS increases from 5.5 KPa to 0.39 MPa, showing typical rate-dependent characteristics [62].

In addition, the three-dimensional network structure of the hydrogel gives it excellent flexibility (up to more than 1000% strain), which can perfectly fit the complex surface of the skin (such as joints, corners, etc.), avoiding the mechanical mismatch problem of traditional rigid materials. The modulus of the hydrogel is close to that of human soft tissue (0.1–100 kPa), reducing mechanical irritation to the skin. Some hydrogels, such as dual-network hydrogels, can be fixed directly to the skin through physical/chemical adhesion without additional adhesives. Therefore, hydrogels make a good material for manufacturing substrates. Wang et al. proposed a hydrogel-based E-skin capable of dual-mode temperature and strain sensing [68]. The composite hydrogel synthesized by the freeze–thaw method has remarkable flexibility, ductility and adaptability to human tissues. The hydrogel-based E-skin is used on various parts of the human body, including the cheeks, head, feet, fingers and elbows. The results show that hydrogels have broad application prospects in monitoring the neck posture and activity state of sedentary office workers.

**Figure 1 micromachines-16-00394-f001:**
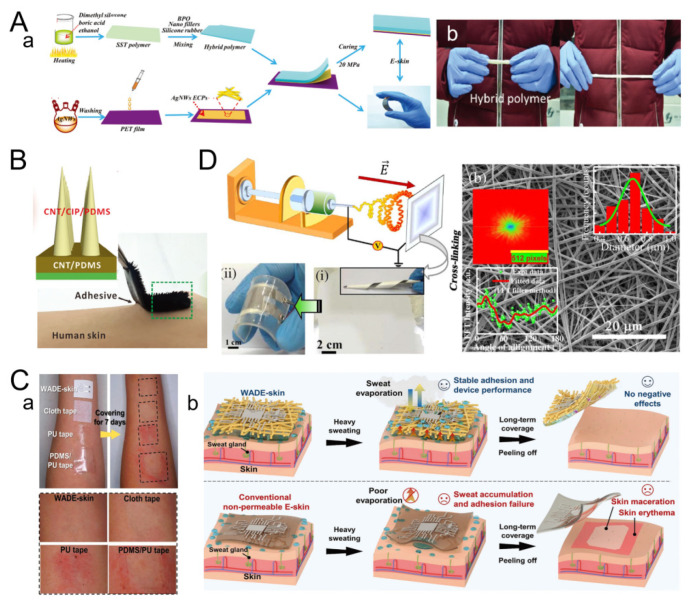
(**A**) Preparation process of hybrid polymers. (Reproduced with permission from the Reference [62]). (**B**) Schematic diagram of the E-skin with microcilia. Consisting of three parts: conducting CNT/CIP/PDMS microcilia, conducting CNT/PDMS substrate and bonding layer. (Reproduced with permission from the Reference [67]). (**C**) Skin irritation results on forearm covered with WADE-skin, cloth tape, PU tape and PDMS/PU tape. (Reproduced with permission from the Reference [19]). (**D**) Device diagram of GNFs prepared by crosslinking method. (Reproduced with permission from the Reference [69]).

#### 2.1.2. Nanofiber Fabric

Traditional polymer films are difficult to adapt to the moist microenvironment of the skin. Due to the impermeability of most elastic membranes, it not only prevents the skin from sweating, but is also difficult to adhere to wet skin. As a result, most types of E-skin have a hard time working underwater.

The multi-layer interleaved nanofiber fabric contains a large number of micro-nano-grade pores, providing a channel for water-heat transmission, which helps avoid discomfort caused by local skin temperature and humidity increase when worn for a long time. For example, Chen et al. reported a wet-adaptive E-skin (WADE-skin) with a substrate composed of PAAND fiber pads and TPU fiber gaskets (Figure 1C) [19]. This WADE-skin also has excellent stretchability, wet adhesion, permeability, biocompatibility and water resistance, which allows it to adhere quickly after a few seconds of contact with human skin, and it can remain stable for several weeks even under wet conditions, without any negative effects on skin health.

In addition, Ghosh et al. made bionic piezoelectric E-skin (Figure 1D) from structurally stable fish gelatin nanofibers (GNFs) based on electrospinning technology. Real-time monitoring of human physiological signals with non-invasive rational strategies due to excellent mechanical sensitivity and fatigue resistance (over 6 months) [69].

### 2.2. Interconnection Materials

#### 2.2.1. Silver Nanowires (AgNWs)

The metal has high conductivity, so the metal film is used as a conductive material for E-skin [70,71,72]. However, the rigidity of the metal makes its tensile properties worse. Using metal nanowires or metal nanosheets as fillers can ensure the stretchability of the E-skin while conducting electricity. Using AgNWs (or silver nanosheets) as fillers for elastomeric polymers can maintain good stretchability and electrical conductivity under large strains exceeding the tensile limit of human skin by 30% [73,74,75,76]. In addition, the diameter and spacing of AgNWs are much smaller than the wavelength of visible light, allowing light to penetrate. Therefore, AgNWs are an ideal filling material.

Wang et al. designed and tested a sensor array built with an improved sandwich structure. The upper and lower layers are PDMS films embedded with conductive bands, which are formed by a network of PDMS-based AgNWs covered with nanoscale metal films. Wang et al. investigated its static and dynamic characteristics; the result showed that the sensor can detect contact pressure at a tensile rate of 30%, with low hysteresis and good linearity [77]. Using AgNW as an electrode and an ultra-thin colorless polyimide (cPI) layer as substrate, Won et al. prepared an E-skin with high light, thermal, electrical properties and biocompatibility based on the kirigami method (Figure 2A). The electrode can be stretched from a small strain of 0% to an ultra-high strain of 400% with negligible mechanical lag and good strain capacity even after 10,000 stretches [78]. It can be seen that AgNWs are a promising next-generation flexible electronic device. However, the current AgNWs have the disadvantages of high contact resistance and poor long-term stability. To combat the loss of electrical conductivity in AgNWs, researchers have tried to add a protective layer between the AgNWs and the substrate. For example, Chen et al. used AgNWs as the conductive layer and polyvinyl alcohol (PVA) as the interfacial protective layer to develop a flexible and highly conductive nanocomposite film, as shown in Figure 2B. Studies have shown that the stability of AgNWs is greatly improved with a protective layer [79].

#### 2.2.2. Carbon Nanomaterials

Carbon nanomaterials (carbon black, carbon nanotubes, graphene, graphene oxide) are another common conductive material [44,80,81]. Among them, carbon nanotubes (CNTs) are widely used as conductive materials for E-skin due to their excellent electrical conductivity and stability. The one-dimensional structure of CNTs ensures good stretchability of E-skin, and they are often used as a filler for elastomeric polymers. Yamada et al. made stretchable E-skin from neatly arranged single-walled carbon nanotube films. When subjected to tension, the nanotube film breaks into a gap. This mechanism allows the film to act as a strain sensor capable of measuring up to 280% of the strain. The carbon nanotube sensor is mounted on stockings, bandages and gloves to create an E-skin that can detect different human movements, such as typing, breathing and speaking [82].

An et al. proposed a highly flexible non-contact E-skin based on femtosecond laser direct writing (FsLDW), as shown in Figure 2C [83]. The photoreduced GO pattern is used as the conductive electrode, and the original GO film is used as the sensing layer. The prepared E-skin has high sensitivity, rapid response and recovery behavior, good long-term stability and excellent mechanical robustness.

Fu et al. designed a sinusoidal shape of high-performance carbon fiber (CF) as the electrode material to give the E-skin a stress–strain curve similar to that of human skin (Figure 2D). By controlling the content and shape of the CF electrode, the Young’s modulus, tensile strength and failure strain of the E-skin can be easily adjusted to match the Young’s modulus, tensile strength and failure strain of human skin. The E-skin prepared based on the electrode demonstrated spatio-temporal recognition of the position and magnitude of the contact force when monitoring the dynamic stress distribution [84].

**Figure 2 micromachines-16-00394-f002:**
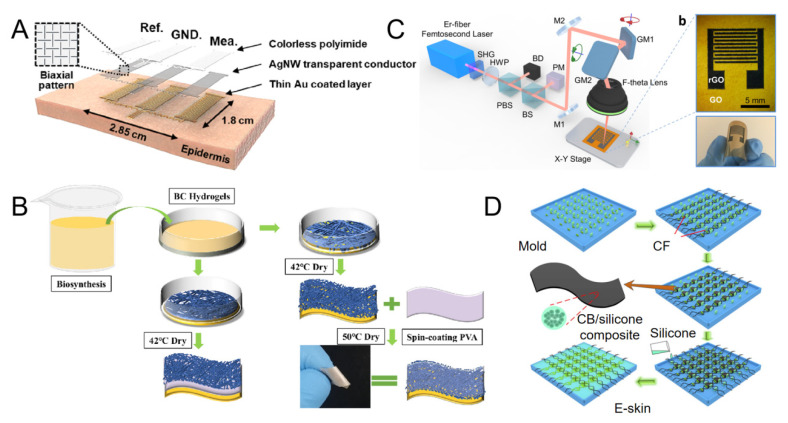
(**A**) Schematic diagram of the transparent kirigami EP sensor. (Reproduced with permission from the Reference [77]). (**B**) Fabrication of nanocomposite film. (Reproduced with permission from the Reference [79]). (**C**) FsLDW system schematic. (Reproduced with permission from the Reference [83]). (**D**) Schematic diagram of E-skin production process: 3D printed mold with microcolumn array. The CF electrode is regularly wound around the microcolumn in one direction. Surface of the disk silicone rubber/silicone rubber composite is coated with uncured CB/silicone resin to carry the CF electrode and obtain the E-skin. (Reproduced with permission from the Reference [84]).

#### 2.2.3. Liquid Metal (LM)

The application and exploration of LMs in E-skin are rapidly increasing, which stems from their high thermal and electrical conductivity. LM, either in bulk form or in granular form, offers superior stretchability, which is a good material for making super-malleable E-skin. Among the numerous types of LMs, LM gallium offers unique properties. First, compared to the LM mercury, LM gallium has an extremely low vapor pressure at room temperature and very little water solubility, so it is much safer than amalgam, which is more suitable for the medical field. In addition, the two-dimensional oxide formed on the liquid gallium alloy’s surface can be used as a microchannel of the LM, increasing its wettability. Yang et al. proposed a gallium LM-based triboelectric nanogenerator (LM-TENG) with Galinstan as the electrode and silicone rubber as the triboelectric and packaging layer (Figure 3A). The LM-TENG maintains stable performance under various deformations, such as stretching, folding and twisting. The lump-like, bangle-shaped LM-TENG captures mechanical energy from human movement (walking, arm shaking or clapping) to sustainably drive wearable electronics [85]. In addition, Ozutemiz’s research found that the HCl-vapor treatment significantly increased the LM interface’s conductivity, while increasing the strain limit of the soft circuit and the reproducibility of the interface (Figure 3B) [86].

#### 2.2.4. Pure Metal

Metals such as gold (Au), silver (Ag), copper (Cu) and aluminum (Al) are the most useful conductors because of their high electrical conductivity. The electrical conductivity of Ag, Cu and Au is much higher than that of carbon materials (such as graphene) and conductive polymers. They are highly ductile and withstand 10–50% strain without breaking. Inert metals such as gold and platinum (Pt) are stable and non-toxic in biological environments and are suitable for implantable devices such as neural interface electrodes.

However, the elastic modulus of the metal is much higher than that of human skin and polymer substrate, resulting in stress concentration at the interface, and the device is prone to failure after repeated bending, which cannot meet the needs of high-tensile E-skin. It is not only opaque, but also expensive. Therefore, the application of metal in E-skin is limited.

### 2.3. Sensing Material

The key factor that makes the sensor versatile is the material used. So far, several materials have been used to manufacture multifunctional sensors, mainly including carbon materials, semiconductors, metal oxides and so on.

#### 2.3.1. Carbon Materials

Carbon-based materials have outstanding advantages in conductivity, low toxicity, chemical stability and easy functionalization. Therefore, in addition to being a conductive material, carbon materials are also used to make E-skin’s multifunctional sensors. Carbon materials include allotropes of carbon (carbon nanotubes CNTs, graphene GO, graphite, etc.) and carbides (such as titanium carbide TiC, SiC, MXene, etc.). Sensors based on these carbon materials can sense multiple stimuli simultaneously under long-term wear.

CNTs are one of the ideal sensing materials because of their excellent electrical conductivity, mechanical strength, piezoresistive properties and high flexibility, which have made great contributions to the research of flexible sensors in recent decades. CNTs include single-walled carbon nanotubes (SWCNTs) and multi-walled carbon nanotubes (MWCNTs). CP et al. used thermoplastic polyurethane as the substrate and MWCNTs as the sensing material to make E-skin with a temperature-sensing function. The E-skin’s thermal sensor has a sensitivity of up to 0.947%. This research opens up new avenues for developing the next generation of E-skin for human–machine interfaces (Figure 3C) [87]. In addition, Xie et al. found that the inclusion of microstructures improved the sensitivity of the material. Cellulose was dissolved in NaOH/urea aqueous solution and combined with CNTs to prepare regenerated cellulose (RC)–CNT composite film. Based on the composite membrane group, E-skin sensor was prepared. The experimental results showed that the RC-CNT composite films had good flexibility, sensing properties and response properties. In vitro toxicity tests showed that the composite membrane had no cytotoxicity. Therefore, flexible sensors based on RC-CNT composite films have promising applications in wearable health monitoring [88].

As a new type of two-dimensional carbon nanomaterial, GO has good electrical properties, high thermal conductivity, chemical stability and good mechanical strength. GO and its reduced form, reduced graphene oxide (rGO), are mass-produced by stripping the material from graphite. Their surface functional groups, including hydroxyl, carboxyl and epoxy groups, are highly sensitive to environmental conditions, including humidity, chemicals and temperature, which are often used to build E-skin. E-skin based on GO is simple to manufacture and has good mechanical stability when stretched, squeezed or twisted. The use of GO in E-skin paves the way for a transparent and versatile sensor. Ho et al. developed a transparent, stretchable, multifunctional E-skin based on GO, as shown in Figure 3D. The simple lamination process allows the E-skin to integrate three functional sensors, humidity, heat and pressure sensors, of which GO and rGO are the sensing materials for humidity and temperature sensors, respectively. Each sensor responds to its associated external stimulus, but is insensitive to the other two stimuli. In addition, all the sensors work simultaneously and are able to report different stimuli individually [81]. Breathable and stretchable E-skin based on CNTs/GO/GelMA was prepared by Li et al. for assessing wound status, as shown in Figure 3E. The structure has good porosity, nanofiber structure and excellent air permeability. In addition, the prepared E-skin matched the mechanical properties of human skin. The moisture-sensing pad has high efficiency in monitoring and early warning of wound interstitial fluid outflow. This study has potential applications in wound management and home medical diagnosis [89].

Due to the lack of timely and portable monitoring and alerting measures, cardiovascular problems and diseases such as sleep apnea cause a large number of deaths each year. Various wearable devices for health monitoring have been extensively studied to reduce mortality. However, the devices themselves can only detect physiological signals, but not sound the alarm. As a result, they must rely on mobile phones or other peripheral devices such as speakers or vibrating motors to sound the alarm, which can cause patients to miss out on optimal treatment. Chen et al. developed a health monitoring device capable of both physiological signal detection and sound alarm, as shown in Figure 3F. A one-step E-skin based on laser-induced graphene (LIG) was found to have good mechanical and acoustic properties for simultaneous health monitoring and alarm. The E-skin has ultra-high sensitivity and can detect various biological signals such as wrist pulse and breathing. An alert can be issued when an anomaly is detected. This study examines the issue of multifunctional sensor integration, further opening up the application of wearable sensors in the field of healthcare [49].

**Figure 3 micromachines-16-00394-f003:**
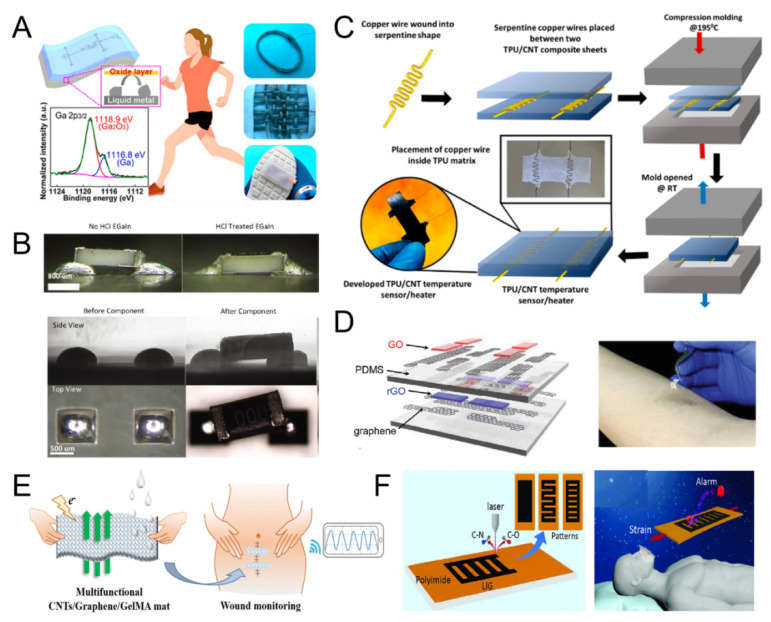
(**A**) Wearable E-skin based on liquid metal. (Reproduced with permission from the Reference [85]). (**B**) HCl treatment on self-aligning. (Reproduced with permission from the Reference [86]). (**C**) Fabrication of E-skin based on TPU/CNT composite material. (Reproduced with permission from the Reference [87]). (**D**) Schematic of a multi-modal E-skin capable of mapping 3 separate stimuli. (Reproduced with permission from the Reference [81]). (**E**) Fabrication of E-skin based on CNTs/graphene/GelMA composites. (Reproduced with permission from the Reference [89]). (**F**) Fabrication of LIG-based E-skin and its application for detecting breath and alerting. (Reproduced with permission from the Reference [49]).

#### 2.3.2. Metal Oxides

Inorganic semiconductor materials, such as ZnO and ZnS, have shown broad application prospects in the field of wearable flexible electronic sensors due to their excellent piezoelectric properties.

Due to its excellent piezoelectric effect and mechanical properties, one-dimensional ZnO nanostructures organized in regular array form have been considered as one of the most promising sensing materials in E-skin. Ghosh et al. reviewed recent advances in one-dimensional ZnO nanostructured arrays for multifunctional piezoelectric sensors [90]. This study provides guidance for the future research direction of E-skin based on one-dimensional ZnO nanostructures, which will accelerate the development of multi-power sensors, human–machine interface chips and self-powered systems. Panth et al. demonstrated a highly sensitive strain sensor based on ZnO with a strain coefficient of up to 1250. The sensitivity of the sensors based on one-dimensional ZnO nanostructures introduced in this study was improved by a factor of 7 [91].

In addition, the surface ends of ZnO have been shown to efficiently functionalize linking molecules such as Disulfide (succinimide propionic acid) (DSP) and (3-amino-propyl) triethoxy-silane (APTES), thereby establishing biological immunoassays with higher sensitivity to specific physiological signals. ZnO treated by pulsed laser deposition (PLD) is often used for the detection of glucose and cortisol. Munje et al. used ZnO films as a functional material for E-skin to detect cortisol in human sweat [92]. Hybrid ZnO has been shown to have the ability to interact with the human body. But the limitations of biocompatible power supplies, high-resolution displays and esthetically compliant designs remain. The recent trend is for free-form displays that can be stretched and embedded in bendable textiles for real tactile sensations while maximizing throughput and minimizing costs, which will change the way E-skin devices interact. Lah et al. analyzed the current status of 3D free-form displays based on mixed ZnO films, analyzed the technical challenges of mixed ZnO films, and anticipated the next generation of surface thin film electronics [93]. He et al. proposed a flexible self-powered E-skin with multiple functions such as perception and detection. Piezoelectric PVDF and tetrapod ZnO (T-ZnO) nanostructures were hybridized on a flexible fabric substrate to obtain tactile sensing behaviors, such as detecting elbow bends or finger presses. This novel material system facilitates the development of flexible, self-powered, multifunctional E-skin [94].

In addition, brittle semiconductor materials can be endowed with flexible characteristics through strain-engineered designs, including wrinkle structures, wavy structures and kirigami architectures [95,96,97,98]. After acquiring the flexible properties, semiconductor materials’ stretchability and fibrous properties have far-reaching applications in sensing and computational electronics in biological and complex environments. Liu et al. created a neuromorphic transistor with stretchable nanowire channels that can sense tactile and visual information and simulate neuromorphic processing capabilities [99]. The device, which is compatible with skin deformation, connects the device to a neuromorphic information processing unit on the finger that can be used for gesture recognition. This work demonstrates the idea of a multi-sensory artificial nerve and neuromorphic system.

## 3. Sensing Mechanisms

Flexible strain/pressure sensors, which measure changes in electrical signals caused by mechanical deformation or pressure stimulation, have generated a flurry of research in recent years due to their potential applications in medical health monitoring, soft robotics and artificial intelligence. Strain/pressure sensors are generally divided into four types according to their working mechanism: piezoelectric, piezoresistive, triboelectric and capacitive (Table 2). The conduction mechanism of strain/pressure sensors is the change in voltage, resistance, current and capacitance caused by the mechanical force of piezoresistive, capacitive, piezoelectric and triboelectric sensors.

### 3.1. Piezoelectric Sensing

E-skins based on piezoresistive and capacitive sensing mechanisms require an external power source to operate, making the system difficult to miniaturize. Piezoelectricity is a mechanism that can directly generate electrical signals and is commonly used in sensors in E-skin. The piezoelectric effect refers to the phenomenon that piezoelectric (PZT) materials produce electrical polarization under mechanical stress (positive piezoelectric effect), or deformation under electric field (inverse piezoelectric effect). The positive piezoelectric effect is mainly used in the E-skin. When the material is subjected to pressure or strain, the internal electric dipoles are oriented, the surface accumulates charge, and the charge signal is collected through the electrode to realize pressure sensing.

The voltage/charge output of PZT materials is proportional to the strain rate and is sensitive to rapidly changing pressures (such as tactile transient responses, vibration signals) with response times up to microseconds. The absence of batteries or external power supplies makes it possible for E-skin based on piezoelectric sensing mechanism to achieve long-term monitoring. However, weak charge signals require charge amplifiers with high input impedance, thus increasing the power consumption of the system. In addition, the preparation costs of high-performance PZT materials are high.

### 3.2. Piezoresistive Sensing

The piezoresistive effect refers to the phenomenon where the material’s resistance changes with mechanical strain. Sensing in the E-skin is achieved by strain-induced changes in material resistivity or geometric size (length, cross-sectional area). Piezoresistive sensors are widely used because of their simple device design, fabrication process and reading method. They are able to convert force changes into resistance changes, which can be easily detected by electrical measurement systems. Many studies are based on piezoresistive mechanisms that integrate conductive nanomaterials into flexible substrates. The researchers found that ultra-thin gold nanowires (two nanometers in diameter and tens of micrometers in length) have high mechanical flexibility and are very effective in creating sensitive sensors. The high electrical conductivity and unique microstructure of the gold nanowires allow the sensors to detect tiny forces. In addition, the stretchability and sensitivity of piezoresistive sensors can be further improved by using conductive porous materials with excellent electrical and mechanical properties. By designing the microstructure and pore size of the porous structure, there is more freedom in the performance control.

Piezoresistive sensors have the advantages of simple structure, relatively low cost and ease of implementation, which makes them more attractive in a variety of applications. However, the sensitivity is limited and is greatly affected by environmental factors such as temperature.

### 3.3. Triboelectric Sensing

The triboelectric effect is based on electron transfer (i.e., triboelectrification) and electrostatic induction during contact separation of the material. In the E-skin, the negative and positive materials are separated after contact, creating a potential difference between the electrodes and converting external mechanical signals into electrical signals. The potential can vary with changes in contact conditions, such as contact time and contact area, which are controlled by external mechanical interactions and can therefore be used as sensing.

Triboelectric sensors can be integrated with piezoresistive sensors and, by using a porous structure, ultimately increase sensitivity and enable more functions and applications. Triboelectric sensors do not require an external power source, and have ultra-high sensitivity and a wide response range. In addition, the advantages of simple structure and low cost make triboelectric sensors widely used. However, it has the disadvantages of lack of static response and environmental sensitivity.

### 3.4. Capacitive Sensing

Capacitance is a parameter that measures charge storage capacity, which is a function of dielectric constant, electrode area and the distance between two electrodes. The capacitance formula is C = ε_0_ε_r_A/d (C is the capacitance, ε_r_ is the dielectric constant, A is the electrode area and d is the spacing). The external force changes the distance or area between the two plates, thereby changing the capacitance. For example, when the film is stretched, d decreases and A increases under the effect of external force, causing C to change. Sensing is achieved in the E-skin by measuring changes in capacitance.

Capacitive sensors have high sensitivity and fast response speed and are sensitive to small pressure changes. However, they are vulnerable to electromagnetic interference, and the production process has higher requirements.

## 4. Applications

### 4.1. Disease Diagnosis

Point-of-Care Testing (POCT) is a rapid diagnostic technology that is carried out on the patient’s side. It uses portable devices and analytical instruments to quickly obtain test results at the sampling site or near the patient, thereby helping doctors make timely diagnosis and treatment decisions. Based on the advantages of low cost, speed, ease of use and independence from instruments, the POCT method is used to monitor health status in non-laboratory environments where there is a lack of professional and technical personnel and perfect laboratory facilities. The common diagnostic methods of POCT include a blood glucose test or urine test, which patients can also conduct at home. It has the advantages of simple operation, low price and no need for formal training. However, it has the disadvantages of low sensitivity, poor accuracy and inability to continuously monitor the status. Therefore, a large number of studies have been devoted to creating next-generation POCT diagnostic tools to overcome the above shortcomings to meet the needs of daily physiological monitoring.

Flexible substrates with enhanced physical and chemical properties have been increasingly integrated into E-skins to detect target biological signals. The E-skin of the sensor is integrated, which not only improves the sensitivity of the detection but can also carry out non-invasive continuous monitoring of physiological signals. With the introduction of the signal acquisition module and data analysis module, POCT diagnosis can carry out signal acquisition and data analysis independently without special equipment. In addition, the integration of wireless communication technology and back-end cloud services has created unprecedented opportunities for real-time and continuous acquisition of physiological signals, which will contribute to the early diagnosis of chronic diseases such as cancer or diabetes.

However, the key limitation of E-skin for continuous, long-term monitoring is that the recorded data cannot be stored in the memory module and the corresponding treatment cannot be triggered based on the stored data. In this regard, Yang et al. systematically evaluated the capabilities of electronic skin (E-skin) devices for health monitoring, focusing on their potential to enable prolonged and precise physiological signal acquisition. Yang et al. comprehensively outlined key detectable biomarkers in the field, including vital signs (heart rate, blood pressure), biochemical parameters (glucose, lactate) and tissue status (wound healing, hydration). By analyzing representative case studies, the review highlighted innovative applications of E-skin in wearable diagnostics, such as continuous sweat analysis, non-invasive blood glucose monitoring and real-time epidermal hydration mapping [16]. Ghosh et al. designed an E-skin with excellent mechanical sensitivity using structurally stable fish gelatin nanofibers (GNFs) (Figure 4A). It can simulate human space–time perception and monitor human physiological signals in a non-invasive way in real time. Furthermore, nano-scale iron and piezoelectric materials in GNFs enable the bio-E-skin to provide self-powered electrical performance, with excellent operational stability of more than 108,000 cycles and fatigue resistance of more than 6 months, solving the problem of external power supply complexity [69].

Cardiovascular diseases (such as coronary heart disease, myocardial infarction, arrhythmia, etc.) and sleep apnea (such as obstructive sleep apnea hypopnea syndrome, OSAHS) are two diseases that seriously affect human health, causing a large number of deaths worldwide each year. The high mortality from these diseases is partly due to a lack of real-time surveillance and alert measures. Wearable devices for health monitoring have been extensively studied to reduce mortality. However, most of the devices themselves can only detect physiological signals, not emit alarms. As a result, devices such as mobile phones, speakers or vibrating motors must be relied upon to sound the alarm, which can cause patients to miss out on optimal treatment. It is very important to develop a wearable health monitoring device which can detect physiological signals and sound an alarm at the same time. Meivel et al. proposed an E-skin that enables continuous observation of human health at home for the early diagnosis and treatment of disease (Figure 4B). The E-skin can store the last three months of data, and the alarm signal alerts a clinician via an IoT controller. This technology provides patients and specialists with continuous examination and real-time data [100]. The E-skin based on laser-induced graphene (LIG) has good mechanical and acoustic properties for real-time monitoring and alarm. The LIG-based E-skin has ultra-high sensitivity and can be used to detect various biological signals such as wrist pulse and breathing. An alert can be issued when an anomaly is detected (Figure 4C). This class of E-skin enables the multifunctional integration required for multiple sensors, further facilitating the application of wearable sensors and healthcare devices [49].

For real-time respiratory monitoring and OSAHS diagnosis, Peng et al. reported a highly sensitive, self-powered E-skin (Figure 4D) based on a triboelectric nanogenerator (TENG). The E-skin consists of multi-layer polyacrylonitrile and polyamide 66 nanofibers as contact pairs and deposited gold as electrodes. It has the advantages of high sensitivity, good stability, good permeability and so on. Therefore, the E-skin also has real-time fine-breath monitoring function. At the same time, a self-powered diagnostic system for OSAHS severity assessment should be further developed to delay its development and improve sleep quality. This study opens up a new approach for real-time respiratory monitoring and clinical detection of sleep-breathing disorders [101].

**Figure 4 micromachines-16-00394-f004:**
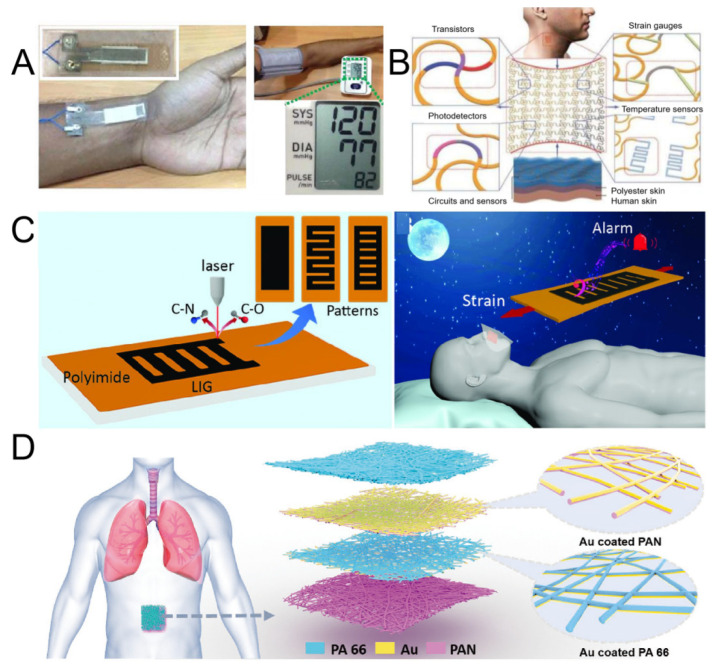
(**A**) E-skin is used to measure wrist pulses. (Reproduced with permission from the Reference [69]). (**B**) Circuit design of E-skin. (Reproduced with permission from the Reference [100]). (**C**) Fabrication process of the E-skin and its use for detecting breath and alerting. (Reproduced with permission from the Reference [49]). (**D**) Schematic diagram of the TENG-based E-skin structure and its application scenario for attaching to the abdomen for respiratory monitoring. (Reproduced with permission from the Reference [101]).

When E-skin detects abnormalities in physiological signals, it can automatically trigger the drug delivery system. This function is based on advanced algorithms and logical judgments to identify precisely when the drug needs to be released, as well as the dose to be released, enabling personalized drug treatment regimens.

The nanopores of silica nanoparticles greatly increase the adsorption surface area of the drug. Son et al. added Resistive Random Access Memory (RRAM) in the system to store data by changing the resistance value. Silica nanoparticles are used as carriers for drug delivery, resistance heaters are used as diffusion acceleration elements, and temperature sensors are used as temperature monitoring elements for controllable transdermal drug delivery. The drug-loaded silica nanoparticles were transferred to the viscous side of the structured polydimethylsiloxane (PDMS) patch. The device can be closely fitted with human skin. The heat generated by the heater degrades the physical binding between the silica nanoparticles and the drug, allowing the drug to be transdermally diffused. The E-skin incorporating sensors and memories can continuously measure physiological signals and then start drug delivery, which is a major advance in the field of personal medicine. This study implies the use of stored information to trigger the onset of treatment [102].

This intelligent drug delivery method not only improves the efficiency of treatment, but also reduces the risk of side effects caused by improper drug use.

### 4.2. Monitoring Physical Health

The technology of E-skin monitoring physiological health has made significant progress in recent years. The human body consists of a number of sensing systems, each of which has its own physiological signal. E-skin is installed in different parts of the body. It can monitor physiological indexes such as heart rate, blood pressure and blood glucose in real time and provide real-time health data for the diagnosis and prevention of diseases (Table 3).

In daily life, continuous monitoring of arterial blood pressure is essential for the prevention and diagnosis of hypertension. However, the existing blood pressure monitoring instruments have the disadvantages of large volume or poor interface performance. In order to overcome the above problems, Li et al. developed a miniaturized E-skin that integrates a piezoelectric sensor array, an active pressure adaptation unit, a signal processing module and an algorithm module [103]. By optimizing the material and sampling system, the interface performance is improved, the monitoring accuracy is greatly improved, and the continuous wireless monitoring of arterial blood pressure is realized (Figure 5A).

Different from the traditional diagnostic methods that require blood sampling, sweat and tear analysis can perform non-invasive monitoring of health status and achieve early diagnosis. Although the monitoring of multiple metabolites has been achieved, the automatic and continuous monitoring of important parameters (sweating rate and electrolyte) in the dehydration process has not been achieved. Honda et al. proposed a wireless, wearable microfluidic sensor system that continuously monitors these parameters in real time, as shown in Figure 5B. The microfluidic sensing system consists of three parts: the top fluid channel for collecting and releasing sweat, the chamber junction for forming and discharging sweat, and the bottom fluid channel with filter paper. In the top fluid channel, a pair of carbon electrodes for impedance sensing and a carbon electrode for sweat rate sensing are integrated. Another carbon electrode for sweat rate sensing is formed in the bottom fluid channel. The amount of sweat is determined by counting and measuring the droplet resistance between the carbon black electrodes in the top and bottom channels. The system also has a Bluetooth module for transmitting data from the sensor to the mobile phone application [104].

Human tears contain many biomarkers (such as glucose, protein, vitamin C, etc.), and monitoring these markers has far-reaching significance for human health. Xu et al. developed a non-invasive E-skin to detect biomarkers in human tears, as shown in Figure 5C. The system is shaped like an eye mask, can be easily worn and has high comfort and good fit. Different sensing areas on the eye mask were modified with specific chromogenic reagents to selectively determine pH, protein, vitamin C and glucose in tears. Only one drop of 20 μL of tears can complete the monitoring, and the reaction is rapid (about 30 s). Subsequently, the smartphone is used to capture and analyze the color signals generated by each sensing area on the eye mask to obtain useful data. The eye mask has great potential in non-invasive detection of tear biomarkers and is widely used in the field of healthcare and clinical monitoring [105].

For the realization of non-invasive monitoring of E-skin, long-term monitoring is essential for disease diagnosis and human–computer interaction. However, hair growth on the surface of human skin can hinder stable contact between the skin and the E-skin, resulting in errors during ultra-long-term monitoring. In order to solve this problem, Tian et al. developed a skin-hair-adaptive viscoelastic dry electrode (VDE), which can bypass the hair, fill the skin wrinkles and improve the stability and durability of the interface impedance (Figure 5D). In electrocardiogram (ECG) monitoring, the experimental results show that VDE can effectively resist hair interference even with severe chest expansion exercise. In addition, VDE can be easily connected to the skull and does not require any EEG cap or bandage, which is an ideal solution for EEG monitoring. This research is a major breakthrough in the field of E-skin for physiological monitoring, overcoming the obstacles of human hair for signal acquisition and processing [106].

Images are usually obtained by hospital MRI. In recent years, wearable imaging technologies such as ultrasound imaging and electrical impedance tomography are essential in modern healthcare monitoring. Based on triboelectric impedance tomography (TIT) technology, Yang et al. developed a new wearable system that enables non-invasive imaging detection of biological tissues [107]. The system’s imaging principle is based on the impedance information collected from the human body’s different soft tissues, and the multi-dimensional impedance data model is constructed to complete the tomography scan of the limbs’ soft tissues. The wearable system can realize the functions of muscle movement observation, movement intention recognition, soft tissue pathological change recognition and so on. Du et al. combined E-skin with ultrasound technology to create a honeycomb breast patch that enables deep scanning and multi-angle image acquisition across the entire breast [108]. This research is the first ultrasound device for breast tissue scanning and imaging, providing a non-invasive method for real-time dynamic monitoring of soft tissue.

### 4.3. Interface with Prosthetics/Computer Systems

Human–computer interaction is a new technology that transmits information between people and electronic devices. In recent years, it has received extensive attention from researchers. The application of wearable devices in prosthetic interfaces is a frontier and developing field. A common interface method is to implant a bionic device into the body of an amputee and permanently integrate it into the bone, similar to the concept of a USB interface. This interface can connect multiple prostheses, reduce customization costs and reduce the need to replace prostheses due to changes in body size. However, this interface can cause trauma to the human body and has the risk of infection. Another interface method is to use muscle signals or neural signals to control prostheses, which has the advantage of non-invasiveness, so this interface method has been widely studied. The traditional surface electromyography (sEMG)-driven upper limb prosthesis control (such as amplitude modulation strategy) has the technical bottleneck of multi-degree-of-freedom synchronous control, especially in the high-order amputation scene of humerus amputation or shoulder joint disconnection. Although pattern recognition (PR) technology can overcome this limitation, its clinical application is limited by the conditions under targeted muscle reinnervation surgery (TMR) surgery. Lauretti et al. proposed a novel transhumerus prosthesis control system that combines EMG signals with multi-modal wearable sensors to achieve multi-degree-of-freedom synchronous control and goal-oriented motion [109]. This scheme breaks through the dependence of TMR surgery and provides a more universal multi-degree-of-freedom control solution for high-level upper limb amputees. In addition, limb position is one of the main factors affecting the sEMG, and the use of inertial measurement unit (IMU) to reflect the overall dynamics of the arm has been used in a complementary way. Compared with sEMG, IMU has the advantages of small size, low cost, no need for skin contact and high signal-to-noise ratio [110]. In recent years, the combination of sEMG and IMU has been the most studied in the literature.

Sensory receptors on the human skin transmit a large amount of tactile and thermal signals from the external environment to the brain. Despite advances in our perception of mechanical and thermal sensations, it is still challenging to replicate these sensory features in prostheses. Recently, intelligent prostheses have been developed using rigid or semi-flexible pressure, strain and temperature sensors to provide a reference for the prosthetic interface of load sensors. Kim et al. demonstrated a smart prosthetic skin integrated with strain, pressure and humidity sensors and temperature sensor arrays for nerve stimulation, as shown in Figure 6(Aa) [111]. These stretchable sensor stimuli contribute to local mechanical and thermal skin-like perception of external stimuli, thus providing opportunities for new prostheses and peripheral nervous system interfaces. The integration of a scalable humidity sensor and a heater enables the regulation of skin humidity and temperature. Then, the corresponding electrical stimulation is transmitted from the prosthesis to the body through the ultra-thin stretchable nanowire electrode of conformal contact to stimulate specific nerves (Figure 6(Ab). In addition, the electrode is decorated with cerium dioxide nanoparticles for controlling inflammation.

Tian et al. introduced an innovative recording device that can fully cover and monitor the electrophysiological signals of the scalp and forearm, as shown in Figure 6B [112]. Its unique filamentary conductive architecture design effectively reduces the influence of RF-induced eddy current, thus solving the problem of incompatibility between traditional equipment and magnetic resonance imaging technology. Studies have shown that for patients who have undergone targeted muscle reinnervation surgery, the device not only supports long-term EEG monitoring and magnetic resonance imaging, but also achieves multifunctional precise control of the transbrachial prosthesis with its large-area interface (Figure 6(Bb)).

Since the 1940s, the electroencephalogram (EEG) has become a non-invasive tool for exploring human brain activity and has been widely used in clinical research and diagnosis. Traditional EEG electrodes are mostly composed of rigid metal disks, which are fixed to the head with a mesh cap and a chin strap. The electrolyte gel can effectively reduce the skin impedance and achieve efficient conduction of electrical signals. However, this configuration is prone to cause skin irritation (such as erythema), and over time, gel drying leads to a decrease in electrical performance, limiting long-term use. In recent years, dry electrode technology has emerged, using needles, probes, capacitive disks, conductive composites or nanowires to replace gels. Although the prospect is promising, it faces challenges such as complex preparation, cumbersome wiring and bulky fixation devices, which hinder the application of EEG in long-term monitoring of neurological dysfunction. For example, although the microneedle electrode can continuously record EEG for several hours, it is difficult to meet the needs of long-term monitoring due to insufficient comfort and ease of use.

By combining a thin reversible adhesive with a capacitive electrode on the scalp surface, although some problems are avoided, the existing design is still limited by the heavy rigid electrode structure, which is prone to mechanical stratification in use, difficult to maintain fixation during bathing and requires additional protection during sleep. Although there are improved versions that try to insert the design through the ear canal, they sacrifice hearing and do not completely overcome other defects of the scalp mounting system.

At present, the latest development of non-invasive EEG electrode technology has opened up a new way for neural disease diagnosis and brain–computer interface data collection. However, due to the stimulation of the skin interface and the irreversible degradation of the electromechanical performance, it is difficult for the existing technology to maintain high efficiency in continuous monitoring for several days. Norton et al. proposed an innovative scheme, which adopts a soft and foldable open fractal grid geometric electrode array, which can closely fit the complex surface of the auricle and mastoid, and realize high-fidelity long-term EEG recording without significant thermal, electrical or mechanical load, providing a new perspective for the future development of EEG technology [113].

## 5. Conclusions

E-skin converts external information into electrical signals to provide bionic perception of human skin. It has the advantages of flexibility, light weight and user comfort. It has attracted great attention in the fields of intelligent robots, human–computer interaction, healthcare monitoring and wearable electronic devices. In this paper, the main manufacturing materials of E-skin are introduced in detail, and the latest progress of its application in physiological monitoring, disease treatment, human–computer interaction and other fields is summarized. Although significant progress has been made in E-skin technology, there are still some challenges, such as high sensitivity, high flexibility, complex manufacturing process, high cost and difficult mass production of E-skin.

Firstly, various parts and functions of E-skin have different requirements for material characteristics, and the selection of composite materials with good comprehensive properties is a good strategy, so the development of a variety of composite materials has become a popular research direction. However, in the case of material composites, there are poor interface properties and uneven dispersion between different materials, which is still a problem to be overcome. Through physical or chemical modification of the surface of materials involved in the composite, the surface roughness, chemical functional groups and other characteristics are changed, so as to enhance the affinity between different materials and improve the interface binding force. Chemical grafting can also be used to introduce specific functional groups on the materials’ surface, so that it can chemically react with another material to form a solid chemical bond connection and improve the interface performance. Compatibilizers can also be introduced into the composite material system. Compatibilizers usually contain groups that have affinity with different materials, interact with one material molecule at one end and interact with another material molecule at the other end, which closely connects different materials and effectively improves interface properties. In addition, a special interface transition layer can be constructed at different material interfaces to form a gradient structure, which makes the performance transition between materials smoother, reduces the interface stress concentration and improves the interface performance.

Secondly, the E-skin health monitoring system integrates many new and advanced functions and technologies, which greatly improves the portability, practicality and intelligence of the system. However, when many functional components are integrated into the E-skin, the signal interference problem is common, which is likely to affect the authenticity of the signal obtained. How to avoid the signal interference between different functions is a topic of great research value. In the manufacturing process of E-skin, low-noise and low-interference electronic components are considered. The module that generates strong signal is separated from the module that is sensitive to the signal, and the physical distance between them is increased to reduce the coupling interference of the signal.

In addition, E-skin is faced with aging and failure of materials during long-term use. At this stage, the E-skin developed in the laboratory has a short service life. Improving the service life of E-skin is a necessary prerequisite for its successful entry into the market and commercialization.

In the future, by optimizing the overall structural design of E-skin, its stability in long-term use will be improved. For multi-layer composite structures, the layers are tightly bound by chemical bonds or physical entanglement to avoid material failure caused by separation between layers. For example, a transition layer is introduced between the sensor layer and the substrate, and the material of the transition layer has good compatibility with the upper and lower layers, which can effectively disperse the stress and prevent the material from cracking and aging due to the concentration of mechanical stress. In addition, the selection of self-healing materials can respond to changes in the environment and adjust the conformation of the molecular chain, so as to achieve self-healing and extend the service life.

In summary, as a new type of sensor that mimics the perception function of human skin, E-skin has broad application prospects in the fields of robotics, medical monitoring and wearable devices. With the continuous progress and innovation of technology, E-skin will bring more convenience and possibility to human lives.

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
