# Peer review of "Wearable Medical Devices: Application Status and Prospects"

_micromachines, 2025, doi:10.3390/mi16040394_

Round 1

Reviewer 1 Report

Comments and Suggestions for Authors

This manuscript provides a systematic review of electronic skin (E-skin) technology in wearable medical devices, focusing on key aspects such as material selection, physiological data acquisition, signal processing, wireless communication, and power management. The review offers valuable insights into the current state of the field and provides a technical framework and inspiration for future research. However, there are still deficiencies in literature coverage and summarization, with insufficient discussion of the latest research findings and comparative analysis of different technical solutions. It is recommended for publication after the following revisions:

  1. The manuscript compares different technologies and methods in a scattered manner. It is suggested to compile these comparisons into a table for a systematic contrast and summary of various materials, technologies, and applications of electronic skin.
  2. The Conclusions section should more specifically highlight the innovative aspects of future research directions.
  3. Some figures are not clear enough. Please check the resolution of Figures 2, 4(B), and 6(B).
  4. In fact, brittle semiconductor materials can be endowed with flexible characteristics through strain-engineered designs, including wrinkle structures, wavy structures, and kirigami architectures. Please provide additional explanations and references (Nature 2013, 499, 458; Nat. Commun. 2012, 3, 770; Nat. Mater. 2015, 14, 785; Small 2021, 17: 1905332).
  5. Further elaboration is needed on the applications of flexible (Nature Communications 2024, 15: 3454; Adv. Sci. 2021, 2102036), stretchable (ACS Nano 2022, 16: 2282-2291), and fiber-shaped (Nano Energy 2022, 104: 107898) electronic devices for sensing and computing in biological and complex environments.
  6. The manuscript uses terms such as “E-skin” and “electronic skin” inconsistently. It is recommended to standardize terminology throughout the manuscript.
  7. The inconsistent use of uppercase (AB) and lowercase (ab) letters in figure labels (e.g., Figure 6A-b) is confusing. Please standardize the use of uppercase letters for labeling.

Author Response

Reviewer 1#

This manuscript provides a systematic review of electronic skin (E-skin) technology in wearable medical devices, focusing on key aspects such as material selection, physiological data acquisition, signal processing, wireless communication, and power management. The review offers valuable insights into the current state of the field and provides a technical framework and inspiration for future research. However, there are still deficiencies in literature coverage and summarization, with insufficient discussion of the latest research findings and comparative analysis of different technical solutions. It is recommended for publication after the following revisions:

  1. The manuscript compares different technologies and methods in a scattered manner. It is suggested to compile these comparisons into a table for a systematic contrast and summary of various materials, technologies, and applications of electronic skin.

Response: We thank the editor for the comment. 

According to editor’s suggestion, we have added three tables for a systematic contrast and summary of various materials, technologies, and applications of electronic skin.. We have marked it in red in the paper.

Table 1.  Advantages and disadvantages of different materials.

Materials

Advantages

Disadvantages

Substrate material

High molecular polymer

-Good flexibility

-Biocompatibility

-Chemical stability

-Limited sensitivity

Nanofiber fabric

-High specific surface area

-Effective adsorption and transfer of substances

-Good flexibility and breathability

-Low mechanical strength

Conductive materials

Silver nanowires (AgNWs)

-Excellent electrical conductivity

-High transparency

-Good flexibility

-Easy oxidation in the air

-High cost

Carbon nanomaterials

-Excellent electrical, mechanical and thermal properties

-High chemical stability

-Poor dispersion

Liquid metal (LM)

-Good liquidity

-Good electrical conductivity and stretchability

-Poor biocompatibility

Sensing material

Carbon materials

-High sensitivity

-Quick response

-High chemical stability

-Poor dispersion

Metal oxides

-Good piezoelectricity, gas sensitivity and optical properties

-Good chemical stability

-Weak electrical conductivity

Table 2. Advantages and disadvantages of different sensing mechanisms.

Mechanisms

Principle

Sensing range

Response speed

Self-Powered

Environmental sensitivity

Advantages

Disadvantages

Piezoelectric sensing

Positive piezoelectric effect

0.1 Pa-100 MPa

μs

Yes

Temperature

-No external power supply required. -Quick. response.

-Weak output signal.

-Limited material. selection.

Piezoresistive sensing

Resistivity/geometric change

0.01%-1000% strain

ms

No

Humidity, temperature

-Simple structure.

-Low cost. -Easy to implement.

-Limited sensitivity.

-Greatly affected by environmental factors.

Triboelectric sensing

Contact electric/electrostatic induction

0.1 N-100 N

ms

Yes

Humidity

-Low cost.

-Self-electricity.

-Unstable charge generation.

-Greatly affected by the friction mode and environmental humidity.

Capacitive sensing

Dielectric constant/distance variation

0.1%-500% strain

μs

No

Temperature

-High sensitivity.

-Fast response speed.

-Sensitive to small pressure changes.

-Susceptible to electromagnetic interference.

-High production process requirements. 

Table 3. Applications, functions, challenges and examples of E-skin.

Applications

Functions

Challenges

Examples

Disease diagnosis

Monitor the concentration of bioelectrical signals and chemical substances in the body to assist in the early diagnosis of diseases.

-Biocompatibility challenges to prevent triggering an immune response.

-Accurate signal interpretation and analysis are complicated.

Implantable cancer marker monitoring chips.

Monitoring physical health

Real-time monitoring of physiological parameters such as heart rate, blood pressure, body temperature, and sweat composition to assess overall health

-Long-term wearing comfort is poor.

-Signals are susceptible to motion and environmental interference.

Smartwatch with an E-skin patch that continuously monitors heart rate and records steps and calories burned.

Interface with prosthetics

Prosthetic limb is endowed with tactile perception, so that the user can feel the grip strength, the surface texture of the object, and improve operation’s accuracy.

-Efficient integration with the human nervous system is difficult.

-Precision of tactile feedback needs to be improved.

Bionic arms with E-skin can accurately sense and respond to the state of grasping objects.

Interface with computer systems

Enhance the interactive experience of electronic products, such as pressure sensing operation for mobile phones and accurate recognition of touch gestures for computers

-Structure integration is difficult.

-High cost.

Some high-end mobile phones use E-skin technology to support heavy pressure and light touch to achieve different functional operations.

  1. The Conclusions section should more specifically highlight the innovative aspects of future research directions.

Response: We thank the editor for the comment.

According to editor’s suggestion, we have highlighted the innovative aspects of future research directions more specifically. We have marked it in red in the paper.

Conclusion

    “E-skin converts external information into electrical signals to provide bionic perception of human skin. It has the advantages of flexibility, light weight and user comfort. It has attracted great attention in the fields of intelligent robots, human-computer interaction, health care monitoring and wearable electronic devices. In this paper, the main manufacturing materials of E-skin are introduced in detail, and the latest progress of its application in physiological monitoring, disease treatment, human-computer interaction and other fields is summarized. Although significant progress has been made in E-skin technology, there are still some challenges, such as high sensitivity, high flexibility, complex manufacturing process, high cost and difficult mass production of E-skin.

Firstly, various parts and functions of E-skin have different requirements for material characteristics, and the selection of composite materials with good comprehensive properties is a good strategy, so the development of a variety of composite materials has become a popular research direction. However, in the process of material composite, there are poor interface properties and uneven dispersion between different materials, which is still a problem to be overcome. Through physical or chemical modification of materials’ surface involved in the composite, the surface roughness, chemical functional groups and other characteristics are changed, so as to enhance the affinity between different materials and improve the interface binding force. Chemical grafting can also be used to introduce specific functional groups on materials’ surface, so that it can chemically react with another material to form a solid chemical bond connection and improve the interface performance. Compatibilizers can also be introduced into the composite material system. Compatibilizers usually contain groups that have affinity with different materials, interact with one material molecule at one end, and interact with another material molecule at the other end, which closely connects different materials and effectively improves interface properties. In addition, a special interface transition layer can be constructed at different material interfaces to form a gradient structure, which makes the performance transition between materials smoother, reduces the interface stress concentration, and improves the interface performance.

Secondly, the E-skin health monitoring system integrates many new and advanced functions and technologies, which greatly improves the portability, practicality and intelligence of the system. However, when many functional components are integrated into the E-skin, the signal interference problem is common, which is likely to affect the authenticity of the signal obtained. How to avoid the signal interference between different functions is a topic of great research value. In the manufacturing process of E-skin, low-noise and low-interference electronic components are considered. The module that generates strong signal is separated from the module that is sensitive to the signal, and the physical distance between them is increased to reduce the coupling interference of the signal.

In addition, E-skin is faced with aging and failure of materials during long-term use. At this stage, the E-skin developed in the laboratory has a short service life. Improving the service life of E-skin is a necessary prerequisite for its successful entry into the market and commercialization.

In the future, by optimizing the overall structural design of E-skin, its stability in long-term use will be improved. For multi-layer composite structures, the layers are tightly bound by chemical bonds or physical entanglement to avoid material failure caused by separation between layers. For example, a transition layer is introduced between the sensor layer and the substrate, and the material of the transition layer has good compatibility with the upper and lower layers, which can effectively disperse the stress and prevent the material from cracking and aging due to the concentration of mechanical stress. In addition, the selection of self-healing materials can respond to changes in the environment and adjust the conformation of the molecular chain, so as to achieve self-healing and extend the service life.

  In summary, as a new type of sensor that mimics the perception function of human skin, E-skin has broad application prospects in the fields of robotics, medical monitoring, and wearable devices. With the continuous progress and innovation of technology, E-skin will bring more convenience and possibility to human beings.”

  1. Some figures are not clear enough. Please check the resolution of Figures 2, 4(B), and 6(B).

Response: We thank the editor for the comment.

According to editor’s suggestion, we have improved the resolution of Figures 2, 4(B), and 6(B).

Figure 2.

Figure 4.

Figure 6.

  1. In fact, brittle semiconductor materials can be endowed with flexible characteristics through strain-engineered designs, including wrinkle structures, wavy structures, and kirigami architectures. Please provide additional explanations and references (Nature 2013, 499, 458; Nat. Commun. 2012, 3, 770; Nat. Mater. 2015, 14, 785; Small 2021, 17: 1905332).

Further elaboration is needed on the applications of flexible (Nature Communications 2024, 15: 3454; Adv. Sci. 2021, 2102036), stretchable (ACS Nano 2022, 16: 2282-2291), and fiber-shaped (Nano Energy 2022, 104: 107898) electronic devices for sensing and computing in biological and complex environments.

Response: We thank the editor for the comment.

According to editor’s suggestion, we have provided additional explanations and references. We have marked it in red in the paper.

2.3.2

“Inorganic semiconductor materials, such as ZnO and ZnS, have shown broad application prospects in the field of wearable flexible electronic sensors due to their excellent piezoelectric properties.

Due to its excellent piezoelectric effect and mechanical properties, one-dimensional ZnO nanostructures organized in regular array form have been considered as one of the most promising sensing materials in E-skin. Ghosh et al. reviewed recent advances in one-dimensional ZnO nanostructured arrays for multifunctional piezoelectric sensors[90]. This study provides guidance for the future research direction of E-skin based on one-dimensional ZnO nanostructures, which will accelerate the development of multi-power sensors, human-machine interface chips, and self-powered systems. Panth et al. demonstrated a highly sensitive strain sensor based on ZnO with a strain coefficient of up to 1250. The sensitivity of the sensors based on one-dimensional ZnO nanostructures introduced in this study was improved by a factor of 7[91].

In addition, the surface ends of ZnO have been shown to efficiently functionalize linking molecules such as Disulfide (succinimide propionic acid) (DSP) and (3-amino-propyl) triethoxy-silane (APTES), thereby establishing biological immunoassays with higher sensitivity to specific physiological signals. ZnO treated by pulsed laser deposition (PLD) is often used for the detection of glucose and cortisol. Munje et al. used ZnO films as a functional material for E-skin to detect cortisol in human sweat[92]. Hybrid ZnO has been shown to have the ability to interact with the human body. But the limitations of biocompatible power supplies, high-resolution displays and aesthetically compliant designs remain. The recent trend is for free-form displays that can be stretched and embedded in bendable textiles for real tactile sensations while maximizing throughput and minimizing costs, which will change the way E-skin devices interact. Lah et al. analyzed the current status of 3D free-form displays based on mixed ZnO films, analyzed the technical challenges of mixed ZnO films, and looked forward to the next generation of surface thin film electronics[93]. He et al. proposed a flexible self-powered E-skin with multiple functions such as perception and detection. Piezoelectric PVDF and tetrapod ZnO (T-ZnO) nanostructures were hybridized on a flexible fabric substrate to obtain tactile sensing behaviors, such as detecting elbow bends or finger presses. This novel material system facilitates the development of flexible, self-powered, multi-functional E-skin[94].

In addition, brittle semiconductor materials can be endowed with flexible characteristics through strain-engineered designs, including wrinkle structures, wavy structures and kirigami architectures[95-98]. After acquiring the flexible properties, semiconductor materials’ stretchability and fibrous properties have far-reaching applications in sensing and computational electronics in biological and complex environments. Liu et al. created a neuromorphic transistor with stretchable nanowire channels that can sense tactile and visual information and simulate neuromorphic processing capabilities[99]. The device, which is compatible with skin deformation, connects the device to a neuromorphic information processing unit on the finger that can be used for gesture recognition. This work demonstrates the idea of a multi-sensory artificial nerve and neuromorphic system.”

  1. The manuscript uses terms such as “E-skin” and “electronic skin” inconsistently. It is recommended to standardize terminology throughout the manuscript.

Response: We thank the editor for the comment.

According to editor’s suggestion, we have changed all “electronic skin” into “E-skin”.  We have marked it in red in the paper.

  1. The inconsistent use of uppercase (AB) and lowercase (ab) letters in figure labels (e.g., Figure 6A-b) is confusing. Please standardize the use of uppercase letters for labeling.

Response: We thank the editor for the comment.

According to editor’s suggestion, we have standardize the use of uppercase letters for labeling and reformated the figures.

Reviewer 2 Report

Comments and Suggestions for Authors

This review summarizes the development of wearable devices from the perspectives of materials and applications. From the reviewer’s standpoint, the manuscript can be further improved in the following aspects:

  1. Since this review is intended for the journal Micromachines, the reviewer recommends a more in-depth discussion on device design and sensing mechanisms throughout the manuscript.
  2. In the section on high molecular polymers, the authors primarily introduce common commercial materials such as PI, PET, and PDMS. To improve the comprehensiveness of the manuscript, the reviewer suggests including a discussion of emerging materials, such as hydrogels. Additionally, the phrase “PDMS hydrogel” in line 96 is inaccurate and should be revised.
  3. The statement in line 116 regarding the impermeability of elastic films is not accurate, as PDMS is known to be permeable to moisture.
  4. An important advantage of silver nanowires that is currently omitted is their high optical transparency when embedded in elastomeric matrices.
  5. The separation between “conductive materials” and “sensing materials” is unclear, as most sensing materials discussed in the review are inherently conductive. The reviewer suggests renaming the “conductive materials” section to “interconnection materials” to better reflect its content.
  6. Both the “conductive materials” and “sensing materials” sections lack a discussion on the use of pure metals in structural designs, such as serpentine geometries, which are widely adopted in soft electronics.
  7. In the applications section, lines 355–360 seem to be directly copied from the cited reference. The reviewer recommends carefully revising this portion.
  8. The subsections on diagnosis and physical health monitoring omit wearable imaging technologies such as optical imaging, ultrasound imaging, and electrical impedance tomography, which are essential in modern healthcare monitoring.
  9. The subsection on human-computer interfaces lack mention of prevalent sensing modalities, including inertial measurement units (IMUs), electromyography (EMG), and ultrasound-based sensors.
  10. To enhance the depth and impact of the review, the reviewer encourages the authors to provide more insightful discussions on the potential challenges in the field and propose future research directions.
Comments on the Quality of English Language

no comments

Author Response

Reviewer 2#

This review summarizes the development of wearable devices from the perspectives of materials and applications. From the reviewer’s standpoint, the manuscript can be further improved in the following aspects:

  1. Since this review is intended for the journal Micromachines, the reviewer recommends a more in-depth discussion on device design and sensing mechanismsthroughout the manuscript.

Response: We thank the editor for the comment.

According to editor’s suggestion, we have added a section to process  a more in-depth discussion on device design and sensing mechanisms. We have marked it in red in the paper.

3

Sensing mechanisms

Flexible strain/pressure sensors, which measure changes in electrical signals caused by mechanical deformation or pressure stimulation, have generated a flurry of research in recent years due to their potential applications in medical health monitoring, soft robotics and artificial intelligence. Strain/pressure sensors are generally divided into four types according to their working mechanism: piezoelectric, piezoresistive, triboelectric and capacitive. The conduction mechanism of strain/pressure sensors is the change of voltage, resistance, current and capacitance caused by the mechanical force of piezoresistive, capacitive, piezoelectric and triboelectric sensors.

3.1 Piezoelectric sensing

E-skins based on piezoresistive and capacitive sensing mechanisms require an external power source to operate, making the system difficult to miniaturize. Piezoelectricity is a mechanism that can directly generate electrical signals and is commonly used in sensors in E-skin. Piezoelectric Effect refers to the phenomenon that piezoelectric (PZT) materials produce electrical polarization under mechanical stress (positive Piezoelectric Effect), or deformation under electric field (inverse piezoelectric effect). The positive piezoelectric effect is mainly used in the E-skin. When the material is subjected to pressure or strain, the internal electric dipoles are oriented, the surface accumulates charge, and the charge signal is collected through the electrode to realize pressure sensing.

The voltage/charge output of PZT materials is proportional to the strain rate and is sensitive to rapidly changing pressures (such as tactile transient responses, vibration signals) with response times up to microseconds. The absence of batteries or external power supplies makes it possible for E-skin based on piezoelectric sensing mechanism to achieve long-term monitoring. However, weak charge signals require charge amplifiers with high input impedance, thus increasing the power consumption of the system. In addition, the preparation cost of high-performance PZT materials are high.

3.2 Piezoresistive sensing

Piezoresistive effect refers to the phenomenon that the material’s resistance changes with mechanical strain. Sensing in the E-skin is achieved by strain-induced changes in material resistivity or geometric size (length, cross-sectional area). Piezoresistive sensor is widely used because of its simple device design, fabrication process and reading method. They are able to convert force changes into resistance changes, which can be easily detected by electrical measurement systems. Many studies are based on piezoresistive mechanisms that integrate conductive nanomaterials into flexible substrates. The researchers found that ultra-thin gold nanowires (2 nanometers in diameter and tens of micrometers in length) have high mechanical flexibility and are very effective in creating sensitive sensors. The high electrical conductivity and unique microstructure of the gold nanowires allow the sensors to detect tiny forces. In addition, the stretchability and sensitivity of piezoresistive sensors can be further improved by using conductive porous materials with excellent electrical and mechanical properties. By designing the microstructure and pore size of the porous structure, there is more freedom in the performance control.

Piezoresistive sensors have the advantages of simple structure, relatively low cost, and ease of implementation, which makes them more attractive in a variety of applications. However, the sensitivity is limited and is greatly affected by environmental factors such as temperature.

3.3 Triboelectric sensing

Triboelectric effect is based on electron transfer (i.e., triboelectrification) and electrostatic induction during contact-separation of the material. In the E-skin, the negative and positive materials are separated after contact, creating a potential difference between the electrodes and converting external mechanical signals into electrical signals. The potential can vary with changes in contact conditions, such as contact time and contact area, which are controlled by external mechanical interactions and can therefore be used as sensing.

Triboelectric sensors can be integrated with piezoresistive sensors, and by using a porous structure, ultimately increase sensitivity and enable more functions and applications. Triboelectric sensors do not require an external power source, which have ultra-high sensitivity and a wide response range. In addition, the advantages of simple structure and low cost make triboelectric sensors widely used. However, it has the disadvantages of lack of static response and environmental sensitivity.

3.4 Capacitive sensing

Capacitance is a parameter that measures charge storage capacity, which is a function of dielectric constant, electrode area and the distance between two electrodes. The capacitance formula is C = εâ‚€εáµ£A/d (C is the capacitance, εáµ£ is the dielectric constant, A is the electrode area and d is the spacing). Based on the applied external force causes a change in the distance or area between the two plates, thereby changing the capacitance. For example, when the film is stretched, d decreases and A increases under the effect of external force, causing C to change. Sensing is achieved in the E-skin by measuring changes in capacitance.

Capacitive sensors have high sensitivity, fast response speed and are sensitive to small pressure changes. However, it is vulnerable to electromagnetic interference, and the production process requires higher requirements.

Table 2. Advantages and disadvantages of different sensing mechanisms.

Mechanisms

Principle

Sensing range

Response speed

Self-Powered

Environmental sensitivity

Advantages

Disadvantages

Piezoelectric sensing

Positive piezoelectric effect

0.1 Pa-100 MPa

μs

Yes

Temperature

-No external power supply required. -Quick. response.

-Weak output signal.

-Limited material. selection.

Piezoresistive sensing

Resistivity/geometric change

0.01%-1000% strain

ms

No

Humidity, temperature

-Simple structure.

-Low cost. -Easy to implement.

-Limited sensitivity.

-Greatly affected by environmental factors.

Triboelectric sensing

Contact electric/electrostatic induction

0.1 N-100 N

ms

Yes

Humidity

-Low cost.

-Self-electricity.

-Unstable charge generation.

-Greatly affected by the friction mode and environmental humidity.

Capacitive sensing

Dielectric constant/distance variation

0.1%-500% strain

μs

No

Temperature

-High sensitivity.

-Fast response speed.

-Sensitive to small pressure changes.

-Susceptible to electromagnetic interference.

-High production process requirements. 

  1. In the section on high molecularpolymers, the authors primarily introduce common commercial materials such as PI, PET, and PDMS. To improve the comprehensiveness of the manuscript, the reviewer suggests including a discussion of emerging materials, such as hydrogels. Additionally, the phrase “PDMS hydrogel” in line 96 is inaccurate and should be revised.

Response: We thank the editor for the comment.

According to editor’s suggestion, we have added a discussion of hydrogel. We have marked it in red in the paper. In addition, the phrase “PDMS hydrogel” in line 96 has been corrected.

2.1.1

“Bending of human joints, such as the wrist, often causes human skin to separate from non-sticky wearable devices, which limits signal acquisition and long-term movement monitoring. In view of this, Dai et al. made an adherent E-skin based on PDMS, which consists of a bonding layer and a response layer, as shown in Figure 1B[67]. Among them, the adhesive layer plays a fixed role to ensure conformal coverage of the interface with the skin, thus maintaining the stability and accuracy of the signal. The microcilia of the response layer are mainly used for the perception of inward/outward curvature. In addition, Wang et al. assembled silver nanowires (AgNWs) and PET films with a mixed shear-reinforced polymer/polydimethylsiloxane (SST/PDMS) matrix to develop a novel multifunctional E-skin with protective and multi-sensing properties (Figure 1A). The hybrid SST/PDMS polymer is structurally stable. When the shear frequency varies from 0.1 to 100 Hz, the storage modulus of SST/PDMS increases from 5.5 KPa to 0.39 MPa, showing typical rate-dependent characteristics[62].

In addition, the three-dimensional network structure of the hydrogel gives it excellent flexibility (up to more than 1000% strain), which can perfectly fit the complex surface of the skin (such as joints, corners, etc.), avoiding the mechanical mismatch problem of traditional rigid materials. The modulus of the hydrogel is close to that of human soft tissue (0.1-100 kPa), reducing mechanical irritation to the skin. Some hydrogels, such as dual network hydrogels, can be fixed directly to the skin through physical/chemical adhesion without additional adhesives. Therefore, hydrogels make a good material for manufacturing substrates. Wang et al. proposed a hydrogel-based E-skin capable of dual-mode temperature and strain sensing[68]. The composite hydrogel synthesized by freeze-thaw method has remarkable flexibility, ductility and adaptability to human tissues. The hydrogel-based E-skin is used on various parts of the human body, including the cheeks, head, feet, fingers and elbows. The results show that hydrogels have broad application prospects in monitoring the neck posture and activity state of sedentary office workers.”

  1. The statement in line 116 regarding the impermeability of elastic films is not accurate, as PDMS is known to be permeable to moisture.

Response: We thank the editor for the comment.

According to editor’s suggestion, we have corrected the statement in line 116. We have marked it in red in the paper.

2.1.2 Nanofiber fabric

“Traditional polymer films are difficult to adapt to the moist microenvironment of the skin. Due to the impermeability of most elastic membranes, it not only prevents the skin from sweating, but also is difficult to adhere to wet skin. As a result, most E-skin have a hard time working underwater.”

  1. An important advantage of silver nanowires that is currently omitted is their high optical transparency when embedded in elastomeric matrices.

Response: We thank the editor for the comment.

According to editor’s suggestion, we have added an explanation of AgNWs. We have marked it in red in the paper.

2.2.1 Silver nanowires (AgNWs)

“The metal has high conductivity, so the metal film is used as a conductive material for E-skin[70-72]. However, the rigidity of the metal makes its tensile properties worse. Using metal nanowires or metal nanosheets as fillers can ensure the stretchability of the E-skin while conducting electricity. Using AgNWs (or silver nanosheets) as fillers for elastomeric polymers can maintain good stretchability and electrical conductivity under large strains exceeding the tensile limit of human skin by 30 %[73-76]. In addition, the diameter and spacing of AgNWs are much smaller than the wavelength of visible light, allowing light to penetrate. Therefore, AgNWs is an ideal filling material.”

  1. The separation between “conductive materials” and “sensing materials” is unclear, as most sensing materials discussed in the review are inherently conductive. The reviewer suggests renaming the “conductive materials” section to “interconnection materials” to better reflect its content.

Response: We thank the editor for the comment.

According to editor’s suggestion, we have renamed the “conductive materials” section to “interconnection materials”. We have marked it in red in the paper.

  • Interconnection materials

  1. Both the “conductive materials” and “sensing materials” sections lack a discussion on the use of pure metals in structural designs, such as serpentine geometries, which are widely adopted in soft electronics.

Response: We thank the editor for the comment.

According to editor’s suggestion, we have added a discussion on the use of pure metals in Section 2.2.4. We have marked it in red in the paper.

2.2.4 Pure metal

Metals such as gold (Au), silver (Ag), copper (Cu), and aluminum (Al) are the most useful conductors because of their high electrical conductivity. The electrical conductivity of Ag, Cu and Au is much higher than that of carbon materials (such as graphene) and conductive polymers. They are highly ductile and withstand 10-50% strain without breaking. Inert metals such as gold and platinum (Pt) are stable and non-toxic in biological environments and are suitable for implantable devices such as neural interface electrodes.

However, the elastic modulus of the metal is much higher than that of human skin and polymer substrate, resulting in stress concentration at the interface, and the device is prone to failure after repeated bending, which cannot meet the needs of high-tensile E-skin. It is not only opaque, but also expensive. Therefore, the application of metal in E-skin is limited.

  1. In the applications section, lines 355–360 seem to be directly copied from the cited reference. The reviewer recommends carefully revising this portion.

Response: We thank the editor for the comment.

According to editor’s suggestion, we have carefully revised this portion. We have marked it in red in the paper.

4.1

“Yang et al. systematically evaluated the capabilities of electronic skin (E-skin) devices for health monitoring, focusing on their potential to enable prolonged and precise physiological signal acquisition. Yang et al. comprehensively outlined key detectable biomarkers in the field, including vital signs (heart rate, blood pressure), biochemical parameters (glucose, lactate), and tissue status (wound healing, hydration). By analyzing representative case studies, the review highlighted innovative applications of E-skin in wearable diagnostics, such as continuous sweat analysis, non-invasive blood glucose monitoring, and real-time epidermal hydration mapping[16]. ”

  1. The subsections on diagnosis and physical health monitoring omit wearable imaging technologies such as optical imaging, ultrasound imaging, and electrical impedance tomography, which are essential in modern healthcare monitoring.

Response: We thank the editor for the comment.

According to editor’s suggestion, we have added a discussion on wearable imaging technologies. We have marked it in red in the paper.

4.2

“Images are usually obtained by hospital MRI. In recent years, wearable imaging technologies such as ultrasound imaging and electrical impedance tomography are essential in modern healthcare monitoring. Based on triboelectric impedance tomography (TIT) technology, Yang et al. developed a new wearable system that enables non-invasive imaging detection of biological tissues[107]. The system’s imaging principle is based on the impedance information collection of human body’s different soft tissues, and the multi-dimensional impedance data model is constructed to complete the tomography scan of the limbs’ soft tissues. The wearable system can realize the functions of muscle movement observation, movement intention recognition, soft tissue pathological change recognition and so on. Du et al. combined E-skin with ultrasound technology to create a honeycomb breast patch that enables deep scanning and multi-angle image acquisition across the entire breast[108]. This research is the first ultrasound device for breast tissue scanning and imaging, providing a non-invasive method for real-time dynamic monitoring of soft tissue.”

  1. The subsection on human-computer interfaceslack mention of prevalent sensing modalities, including inertial measurement units (IMUs), electromyography (EMG), and ultrasound-based sensors.

Response: We thank the editor for the comment.

According to editor’s suggestion, we have added a discussion on inertial measurement units (IMUs) and electromyography (EMG). We have marked it in red in the paper.

4.3

“Human-computer interaction is a new technology that transmits information between people and electronic devices. In recent years, it has received extensive attention from researchers. The application of wearable devices in prosthetic interface is a frontier and developing field. A common interface method is to implant a bionic device into the body of an amputee and permanently integrate it into the bone, similar to the concept of a USB interface. This interface can connect multiple prostheses, reduce customization costs, and reduce the need to replace prostheses due to changes in body size. However, this interface can cause trauma to the human body and has the risk of infection. Another interface method is to use muscle signals or neural signals to control prostheses, which has the advantage of non-invasiveness, so this interface method has been widely studied. The traditional surface electromyography (sEMG) driven upper limb prosthesis control (such as amplitude modulation strategy) has the technical bottleneck of multi-degree-of-freedom synchronous control, especially in the high-order amputation scene of humerus amputation or shoulder joint disconnection. Although pattern recognition (PR) technology can overcome this limitation, its clinical application is limited by the conditions under targeted muscle reinnervation surgery (TMR) surgery. Lauretti et al. proposed a novel transhumerus prosthesis control system that combines EMG signals with multi-modal wearable sensors to achieve multi-degree-of-freedom synchronous control and goal-oriented motion[109]. This scheme breaks through the dependence of TMR surgery and provides a more universal multi-degree-of-freedom control solution for high level upper limb amputees. In addition, limb position is one of the main factors affecting the sEMG, and the use of inertial measurement unit (IMU) to reflect the overall dynamics of the arm has been used as a complementary way. Compared with sEMG, IMU has the advantages of small size, low cost, no need for skin contact and high signal-to-noise ratio[110]. In recent years, the combination of sEMG and IMU has been the most studied in the literature.”

  1. To enhance the depth and impact of the review, the reviewer encourages the authors to provide more insightful discussions on the potential challenges in the field and propose future research directions.

Response: We thank the editor for the comment.

According to editor’s suggestion, we have provided more insightful discussions on the potential challenges in the field and propose future research directions. We have marked it in red in the paper.

Conclusion

    “E-skin converts external information into electrical signals to provide bionic perception of human skin. It has the advantages of flexibility, light weight and user comfort. It has attracted great attention in the fields of intelligent robots, human-computer interaction, health care monitoring and wearable electronic devices. In this paper, the main manufacturing materials of E-skin are introduced in detail, and the latest progress of its application in physiological monitoring, disease treatment, human-computer interaction and other fields is summarized. Although significant progress has been made in E-skin technology, there are still some challenges, such as high sensitivity, high flexibility, complex manufacturing process, high cost and difficult mass production of E-skin.

Firstly, various parts and functions of E-skin have different requirements for material characteristics, and the selection of composite materials with good comprehensive properties is a good strategy, so the development of a variety of composite materials has become a popular research direction. However, in the process of material composite, there are poor interface properties and uneven dispersion between different materials, which is still a problem to be overcome. Through physical or chemical modification of materials’ surface involved in the composite, the surface roughness, chemical functional groups and other characteristics are changed, so as to enhance the affinity between different materials and improve the interface binding force. Chemical grafting can also be used to introduce specific functional groups on materials’ surface, so that it can chemically react with another material to form a solid chemical bond connection and improve the interface performance. Compatibilizers can also be introduced into the composite material system. Compatibilizers usually contain groups that have affinity with different materials, interact with one material molecule at one end, and interact with another material molecule at the other end, which closely connects different materials and effectively improves interface properties. In addition, a special interface transition layer can be constructed at different material interfaces to form a gradient structure, which makes the performance transition between materials smoother, reduces the interface stress concentration, and improves the interface performance.

Secondly, the E-skin health monitoring system integrates many new and advanced functions and technologies, which greatly improves the portability, practicality and intelligence of the system. However, when many functional components are integrated into the E-skin, the signal interference problem is common, which is likely to affect the authenticity of the signal obtained. How to avoid the signal interference between different functions is a topic of great research value. In the manufacturing process of E-skin, low-noise and low-interference electronic components are considered. The module that generates strong signal is separated from the module that is sensitive to the signal, and the physical distance between them is increased to reduce the coupling interference of the signal.

In addition, E-skin is faced with aging and failure of materials during long-term use. At this stage, the E-skin developed in the laboratory has a short service life. Improving the service life of E-skin is a necessary prerequisite for its successful entry into the market and commercialization.

In the future, by optimizing the overall structural design of E-skin, its stability in long-term use will be improved. For multi-layer composite structures, the layers are tightly bound by chemical bonds or physical entanglement to avoid material failure caused by separation between layers. For example, a transition layer is introduced between the sensor layer and the substrate, and the material of the transition layer has good compatibility with the upper and lower layers, which can effectively disperse the stress and prevent the material from cracking and aging due to the concentration of mechanical stress. In addition, the selection of self-healing materials can respond to changes in the environment and adjust the conformation of the molecular chain, so as to achieve self-healing and extend the service life.

  In summary, as a new type of sensor that mimics the perception function of human skin, E-skin has broad application prospects in the fields of robotics, medical monitoring, and wearable devices. With the continuous progress and innovation of technology, E-skin will bring more convenience and possibility to human beings.”
